# Comparative Analysis of Various Plant-Growth-Regulator Treatments on Biomass Accumulation, Bioactive Phytochemical Production, and Biological Activity of *Solanum virginianum* L. Callus Culture Extracts

Hazrat Usman [1], Hasnain Jan [2], Gouhar Zaman [1], Mehnaz Khanum [1], Samantha Drouet [3], Laurine Garros [3], Duangjai Tungmunnithum [3,4,5], Christophe Hano [3,5,*] and Bilal Haider Abbasi [1,*]

[1] Department of Biotechnology, Quaid-i-Azam University, Islamabad 45320, Pakistan; usmanhazrat888@gmail.com (H.U.); gouharkhan4400@gmail.com (G.Z.); maryummehanaz@gmail.com (M.K.)
[2] Institute of Biochemical Sciences, National Taiwan University, Taipei 10617, Taiwan; rhasnain849@gmail.com
[3] Laboratoire de Biologie des Ligneux et des Grandes Cultures (LBLGC), INRA USC1328 Université d'Orléans, CEDEX 2, 45067 Orléans, France; samantha.drouet@univ-orleans.fr (S.D.); laurine.garros@univ-orleans.fr (L.G.); duangjai.tun@mahidol.ac.th (D.T.)
[4] Department of Pharmaceutical Botany, Faculty of Pharmacy, Mahidol University, Bangkok 10400, Thailand
[5] Le Studium Loire Valley Institute for Advanced Studies, 1 rue Dupanloup, 45000 Orléans, France
* Correspondence: hano@univ-orleans.fr (C.H.); bhabbasi@qau.edu.pk (B.H.A.)

**Abstract:** *Solanum virginianum* L. (*Solanum xanthocarpum*) is an important therapeutic plant due to the presence of medicinally useful plant-derived compounds. *S. virginianum* has been shown to have anti-cancer, antioxidant, antibacterial, antiaging, and anti-inflammatory properties. This plant is becoming endangered due to overexploitation and the loss of its native habitat. The purpose of this research is to develop an ideal technique for the maximum biomass and phytochemical accumulation in *S. virginianum* leaf-induced in vitro cultures, as well as to evaluate their potential antiaging, anti-inflammatory, and antioxidant abilities. Leaf explants were grown on media (Murashige and Skoog (MS)) that were supplemented with various concentrations and combinations of plant hormones (TDZ, BAP, NAA, and TDZ + NAA) for this purpose. When compared with the other hormones, TDZ demonstrated the best response for callus induction, biomass accumulation, phytochemical synthesis, and biological activities. However, with 5 mg/L of TDZ, the optimal biomass production (FW: 251.48 g/L and DW: 13.59 g/L) was estimated. The highest total phenolic level (10.22 ± 0.44 mg/g DW) was found in 5 mg/L of TDZ, whereas the highest flavonoid contents (1.65 ± 0.11 mg/g DW) were found in 10 mg/L of TDZ. The results of the HPLC revealed that the highest production of coumarins (scopoletin: 4.34 ± 0.20 mg/g DW and esculetin: 0.87 ± 0.040 mg/g DW) was determined for 10 mg/L of TDZ, whereas the highest accumulations of caffeic acid (0.56 ± 0.021 mg/g DW) and methyl caffeate (18.62 ± 0.60 mg/g DW) were shown by 5 mg/L of TDZ. The determination of these phytochemicals (phenolics and coumarins) estimates that the results of our study on biological assays, such as antioxidant, anti-inflammatory, and antiaging assays, are useful for future cosmetic applications.

**Keywords:** biomass; antioxidant; phytochemicals; coumarins; antiaging; anti-inflammatory

## 1. Introduction

For thousands of years, medicinal plants have been used as a primary resource for health and care due to their wide range of biologically active natural products. Medicinal herbs are nowadays still widely used in both developed and developing countries. Around 70% of the world's population relies on traditional herbal knowledge, and 80% of developing-country populations utilize medicinal plants for medical purposes [1]. A similar trend is observed for cosmetic usage [2]. Due to the introduction of synthetic compounds,

the interest in the use of natural resources declined temporarily in the twentieth century. Several studies have demonstrated that the excessive use of synthetic compounds may have some drawbacks and leads to the increase in the prevalence of certain illnesses [3]. Therefore, the use of natural herbal resources is regaining popularity, in particular as a result of the advancement of contemporary methodologies in plant tissue cultures, phytochemical research, and the use of molecular tools to study biological tests [1].

*Solanum virginianum* L. (*Solanum xanthocarpum*) is a popular medicinal plant that belongs to the genus Solanum of the *Solanaceae* family. It is a perennial plant that lives for about 120 days [4]. This plant is distributed among various countries, such as Ceylon, Middle Asia, Malaya, tropical Australia, and Polynesia, and it is easily available [5,6]. In Pakistan, it grows in wild habitats and is known by the vernacular names of Bhatkatiya, Maraghoni, Warumba, or Kantakri. The phytochemical profiling of *S. virginianum* shows the presence of medicinally important phytochemicals, including caffeic acid, methyl caffeate, scopoletin, and esculetin [6–8]. Its fruits, which are delicious golden berries encircled by an expanded calyx, are widely used in traditional medicine to treat a variety of illnesses. Several biological actions associated with their traditional usage have been reported, including antioxidant, anti-inflammatory, and skin-appearance benefits. A phytochemical analysis of *S. virginianum* extracts indicated the presence of a variety of bioactive secondary metabolites, and specifically of caffeic acid, methyl caffeate, scopoletin, and esculetin [8]. Phenolics, such as caffeic acid and its derivatives (including methyl caffeate), and coumarins, such as scopoletin and esculetin, are biosynthesized from L-phenylalanine via the phenylpropanoid pathway [9–11]. Phenylpropaponoids have been linked to the reduction appearance of degenerative illnesses that develops with aging. They can be used in a variety of ways, including as cosmetic, medicinal, and dietary supplements [12–14]. For this purpose, *S. virginianium* could be considered as an attractive source of valuable phenylpropanoids; however, it is becoming endangered due to natural habitat destruction and overexploitation [15].

As a result, modern and recently established research requires an increase in the production and development of this plant's secondary metabolites. Plant tissue culture is currently a potential method for studying plant physiological activity under regulated chemical and physiological settings [16]. Through in vitro cultures, a maximal amount of essential phytochemicals is obtained in a short time with simple harvesting, and without any climatic constraints [17,18]. The current study emphasizes the establishment of a fast and effective protocol for the valued synthesis of secondary metabolites in the callus culture of *S. virginianum*. For this purpose, various plant-growth-regulator (PGR) concentrations, alone or in combination, were applied on medium (MS) to study their effects on callus development, biomass accumulation, and phytochemical production. The antiaging, anti-inflammatory, and antioxidant properties of all the cultures were also tested to investigate their potential uses in cosmetics.

## 2. Materials and Methods

### 2.1. Chemicals and Reagents

All chemicals and solutions (analytical grade) were obtained from Sigma-Aldrich (Saint-Quentin Fallavier, France).

### 2.2. Seed Germination

*S. virginianum* seeds were obtained from the Quaid-i-Azam University seed bank (Islamabad, Pakistan). To test the viability of the seeds, the water floating technique was utilized. The early viable seeds were sterilized for 30 s with 0.1% $HgCl_2$ (mercuric chloride), before being treated with 70% (*v/v*) ethanol for one minute. To avoid any germination problems, seeds were then washed three times with deionized water. The reported protocol of Ahmad et al. [19] was considered for the inoculation of seeds on Murashige and Skoog (MS) media (Sigma-Aldrich, Saint Quentin Fallavier, France), supplemented with 0.8% (*w/v*) agar and 3% (*w/v*) sucrose (Sigma-Aldrich, Saint Quentin Fallavier, France), used as

gelling agent to solidify and carbon source, respectively. The pH was set at 5.6–5.8 with a pH meter, followed by a routine autoclaving procedure at 121 °C for 20 min.

### 2.3. Callus Culture Establishment

Three distinct PGRs (thidiazuron (TDZ), 6-benzylaminopurine (BAP), and 1-naphthaleneacetic acid (NAA)), alone and in a combination of TDZ + NAA, at various concentrations (0.1, 1, 2.5, 5, and 10 mg/L), were used to examine their influence on *S. virginianum* callus establishment. The leaf explant (0.5 cm$^2$), obtained from 30-day-old plantlets of *S. virginianum*, were inoculated on solid MS media containing various concentrations of the PGRs. For each hormonal concentration, 5 explants were placed in a flask, and tests were performed in 3 biological independent assays. Inoculated flasks were placed in growth room at a controlled temperature of 25 ± 1 °C and a photoperiod of 16/8 h (light/dark). Morphology and induction frequency were examined on a weekly basis. After 3 weeks, all callus cultures were subcultured on fresh MS media containing the respective hormonal concentrations. The fresh (FW) and dry (DW) weights of all cultures were measured after harvesting on the 30th day of the subculture.

### 2.4. Extract Preparation

The previously reported procedure was significantly changed in order to prepare extracts of all treated cultures to assess their phytochemical compositions and biological activities [20]. Callus cultures were collected and dried at 60 °C for one day, before being finely crushed by grinding with mortar and pestle. To make the extract, 100 mg of crushed powder was extracted from each callus and dissolved in 500 µL of methanol. It was then vortexed for 10 min, sonicated for 30 min twice, then centrifuged for ten minutes at 15,000 rpm. The supernatant was isolated to analyze the metabolite contents and perform biological assays.

### 2.5. TPC Determination

Folin–Ciocalteu reagent was used for finding the total phenolic contents (TPCs) in all extracts obtained from treated calli, as described previously [21]. In short, callus extract (20 µL) was added to the FC reagent (180 µL) and was incubated for 5 min, followed by the addition of sodium carbonate solution (90 µL). OD of the reaction mixture was taken with the help of a microplate reader at a wavelength of 630 nm. TPC was expressed in mg GAEs (gallic acid equivalents)/g DW using a standard curve of gallic acid (0–50 µg/mL; $R^2$ = 0.98). The total phenolic production (TPP) was calculated using the following equation:

$$\text{TPP (mg/L)} = \text{TPC (mg/g)} \times \text{biomass production in DW (g/L)}$$

### 2.6. TFC Determination

Total flavonoid contents (TFCs) of all hormonal fortified cultures were investigated by following the protocol described by Ahmad et al. [21], using the aluminum chloride (AlCl$_3$) colorimetric method. The volume of the well (96-well microplate) was filled by adding 20 µL of the extracted sample, 10 µL potassium acetate (1 M), 10% *w/v* AlCl$_3$ (10 µL), and 160 µL of ultrapure water. The mixture in the microplate was incubated at 25 °C for 30 min, and absorbance was taken at a 415 nm wavelength with a microplate reader. TFC was calculated in mg (quercetin equivalent)/g DW using a standard curve (0–40 µg/mL quercetin; $R^2$ = 0.998). The total flavonoid production (TFP) was calculated using the following equation:

$$\text{TFP (mg/L)} = \text{TFC (mg/g)} \times \text{biomass production in DW (g/L)}$$

### 2.7. High-Performance Liquid Chromatographic Analysis

Separation and quantification of coumarins and phenolic compounds were accomplished by HPLC. The HPLC system (Agilent Technology, Les Ullis, France) consisted of Varian Prostar (composed of Prostar-230 pump, Prostar-335 array detector, and Prostar-410

autosampler), controlled with Galaxie software (version 1.9.3.2) (Agilent Technology, Les Ullis, France). The earlier reported protocol was followed for isolation of phytochemicals from sample mixture [22]. The mobile phase consisted of two solvents: Solvent A was $H_2O$ acidified with 0.2% (*v/v*) of acetic acid (pH: 2.1,) and Solvent B was methanol. One-hour HPLC separation was obtained with the following linear changes: 0 min (A: 92%; B: 8%), 11 min (A: 88%; B: 12%), 17–25 min (A: 70%; B: 30%), 28 min (A: 67%; B: 33%), 30−35 min (A: 0%; B: 100%), and 36–60 min (A: 92%; B: 8%), at a flow rate of 1 mL/min. Hypersil PEP 300 (C18 column; 250 × 4.0 mm; internal diameter: 5 µm; Thermo Fischer Scientific, Les Ullis, France) was used. For phytochemicals analysis, the temperature was adjusted at 35 °C, and the process was equilibrated for 10 min after each individual run. Scopoletin, esculetin, and caffeic acid were identified and quantified by using authentic standards from Sigma-Aldrich (Saint Quentin Fallavier, France), whereas methyl caffeate was quantified by using a commercial standard from LGC Standard (Molsheim, France) concerning their retention time. The analytical range of calibration curves was from 10–1000 µg/mL in 5 replicates (6 points), and analytical characteristics were scopoletin (y = 7.503x + 0.763; $R^2$ = 0.9991; LOD = 2.9 ng; LOQ = 10.0), esculetin (y = 7.573x + 0.129; $R^2$ = 0.9993; LOD = 2.8 ng; LOQ = 9.9 ng), caffeic acid (y = 6.583x + 0.968; $R^2$ = 0.9991; LOD = 3.1 ng; LOQ = 10.2 ng), and methyl caffeate (y = 6.120x + 1.192; $R^2$ = 0.9996; LOD = 2.2 ng; LOQ = 6.7), with y peak areas against x concentrations. Examination of the samples was performed three times, injected in µg/mL.

### 2.8. Cell-Free Antioxidant Assays

### 2.8.1. DPPH Assay

The DPPH (2,2-diphenyl-1-picrylhydrazyl) free-radical-antioxidant assay was performed for all extracts obtained from treated callus cultures by adopting the protocol of Ahmad et al. [19]. In short, the reaction mixture of 200 µL was made by the addition of 20 µL of methanolic extract to 180 µL DPPH solution, and then the reaction mixture was incubated in the dark for 1 h, and absorbance was recorded by using a microplate reader with reading set at 517 nm. For negative control, a DPPH solution (180 µL mixed with 20 µL DMSO) was used, while ascorbic acid with rates of 5, 10, 20, and 40 µg/mL was used as a positive control. The following formula was applied for the calculation of the DPPH free-radical-scavenging activity:

$$\% \text{ Radical Scavenging activity (RSA)} = 100 \times (1 - AE/AD)$$

where AD denotes the absorption of the reaction mixture, and AE expresses the absorbance of the DPPH solution only at a given wavelength (517 nm).

### 2.8.2. FRAP Assay

The FRAP (ferric antioxidant power) assay was performed using the protocol described by Benzie and Strain [23], with minor modifications. FRAP solution was made by combining 300 mM acetate buffer with 10 mM TPTZ and 20 mM $FeCl_3.6H_2O$. A total of 10 µL of methanolic extract was thoroughly added to 190 µL of FRAP solution and was maintained at room temperature for 15 min. A microplate reader (Multiskan GO, Thermo Fischer Scientific, Illkirch Graffenstaden, France), at a wavelength of 630 nm, was used for the determination of the FRAP value. Antioxidant capacity was expressed in µM TEAC (Trolox C equivalent antioxidant capacity). A blank was prepared with the solvent extract.

### 2.8.3. ABTS Assay

For the ABTS (2,2′-azino-bis(3-ethylbenzothiazoline-6-sulfonic acid) assay, the modified protocol of Tagliazucchi et al. [24], adapted from the microplate [25], was used, with minor modifications. In short, ABTS solution was added in this protocol, and it was prepared by adding 7 mM of ABTS salt and 2.45 mM of potassium persulphate ($K_2S_2O_8$) in an equal ratio for accuracy while incubating for 16 h in dark. The absorbance was determined at 734 nm with a microplate reader (Multiskan GO, Thermo Fischer Scientific, Illkirch Graffenstaden,

France). Methanolic extracts (10 μL) were then added to the solution, and it was again put in the dark for 15 min at 25 °C. Antioxidant capacity was expressed in μM TEAC (Trolox C equivalent antioxidant capacity). A blank was prepared with the solvent extract.

### 2.9. Antiaging Assay

### 2.9.1. Inhibition of AGE (Advanced Glycation End Product) Formation

The protocol reported by Kaewseejan and Siriamornpun [26] was followed to determine the inhibitory potential of all extracts from treated callus cultures of *S. virginianum* towards AGE formulation. The solution mixture was prepared by mixing BSA solution (20 mg/mL) and glucose solution (0.5 M). Then, 0.1 mL phosphate buffer was used to prepare BSA and glucose-solution mixtures, which contained 0.02% (*w/v*) sodium azide (1 mL, pH: 7.4). The extract was prepared by mixing 50 μg/mL DMSO and putting it in an incubator at 37 °C in dark for 5 days. DMSO alone was used as control. The inhibition activity of each sample was measured as percent inhibition by using a fluorescence spectrometer (VersaFluor fluorometer; Saint Quentin Fallavier, France) at 330 nm (excitation) and 440 (emission) wavelengths. Aminoguanidine (150 μM) was used to draw a standard curve for the inhibition of AGE formation.

### 2.9.2. Elastase-Inhibition Assay

Elastase inhibition for all treated cultures was performed according to the protocol of Wittenauer et al. [27] by using porcine pancreatic elastase and *N*-Succinyl-Alanyl-Alanyl-Alanine-*p*-nitroanilide as substrate (Sigma Aldrich, Saint Quentin Fallavier, France). The activity was measured, after the release of *p*-nitroaniline, in percentage inhibition (relative to control). The absorbance was measured at 410 nm by using a microplate reader (BioTek ELX800; Colmar, France). The negative control was prepared by adding solvent mixture without extract (addition of the same volume of extraction solvent (i.e., methanol)). The inhibition activity was measured as the % inhibition corresponding to this negative control. Oleanolic acid (10 μM) was used as standard drug inhibitor for elastase inhibition.

### 2.9.3. Tyrosinase-Inhibition Assay

The protocol of Chai et al. [28] was followed to investigate tyrosinase-inhibition activity for all treated cultures of *S. virginianum* by using (5 mM) L-DOPA (Sigma Aldrich) as a substrate for diphenolase enzyme. In short, 10 μL of the extracted sample with 5 mM of L-DOPA was added by 50 mM of trisodium phosphate buffer (pH: 6.8), making a final volume of 200 μL, and it was mixed with 0.2 mg/mL mushroom tyrosinase enzyme (Sigma Aldrich, Saint Quentin Fallavier, France). The reading was noted by the microplate of the reaction mixture at 475 nm. The negative control was prepared by adding solvent mixture without extract (addition of the same volume of extraction solvent (i.e., methanol)). The inhibition activity was measured as the % inhibition corresponding to this negative control. Kojic acid (10 μM) was used as a standard drug for tyrosinase inhibition.

### 2.9.4. Collagenase-Inhibition Assay

The method of Chai et al. [28] was followed to perform collagenase-inhibition activity for all treated callus extracts, with the assistance of a spectrophotometer and the compound *N*-[3-(2-furyl) acryloyl]-Leu-Gly-Pro-Ala), used as a substrate for Collagenase from the *Clostridium histolyticum* enzyme (Sigma Aldrich, Saint Quentin Fallavie, France). The reduction in the absorbance of the substrate was observed by using a microplate reader (BioTek Instruments, Colmar, France) at 335 nm for 20 min. The negative control was prepared by adding solvent mixture without extract (addition of the same volume of extraction solvent (i.e., methanol)). The inhibition activity was measured as the % inhibition corresponding to this negative control. Phenantroline (100 μM) was used as a standard drug for collegenase inhibition.

### 2.9.5. Hyaluronidase Assay

To estimate the potential of this activity, a hyaluronidase assay was performed by adopting the procedure of Kolakul and Sripanidkulchai [29]. Hyaluronic acid (0.03% (*w/v*)) was used as a substrate for the hyaluronidase enzyme (1.5 units, Sigma Aldrich). A total of 0.1% (*w/v*) (BSA) containing acid albumen solution was used to precipitate undigested hyaluronic acid. The activity was measured at 600 nm by using a microplate reader (BioTek ELX800; Colmar, France). The negative control was prepared by adding solvent mixture without extract (addition of the same volume of extraction solvent (i.e., methanol)). The inhibition activity was measured as the % inhibition corresponding to this negative control. Oleanolic acid (10 μM) was used as a standard drug inhibitor for hyaluronidase inhibition.

### 2.10. Anti-Inflammatory Assays

Anti-inflammatory assays, including cyclooxygenases 1 and 2 (COX1 and COX2), 15-lipoxygenase (15-LOX), and secretory phospholipase (sPLA2), were determined by using the protocol of Shah et al. [30].

### 2.10.1. COX-1- and COX-2-Inhibition Assays

COX2 (human) and COX1 (ovine) kit assays were followed according to the guidelines specified by Cayman Chem (701050; Montluçon; France). Arachidonic acid (1.1 mM) was used as a substrate. The negative control was prepared by adding solvent mixture without extract (addition of the same volume of extraction solvent (i.e., methanol)). The inhibition activity was measured as the % inhibition corresponding to this negative control. Ibuprofen (10 μM) was used as a standard drug inhibitor for both COX1 and COX2. The COX peroxidase enzyme component kit was used for activity measurement, whereas verification of oxidized tetramethyl-*p*-phenylenediamine ($C_{10}H_{16}N_2$) was performed with Synergy II microplate reader (BioTek Instruments, Colmar, France) at 590 nm.

### 2.10.2. 15-LOX-Inhibition Assay

The 15-LOX-inhibition activity of the extracts was determined by using the kit method (Cayman Chem, 760700, Montluçon, France), and by following the guidelines of the manufacturer. Kit was used for the measurement of hydroperoxides generated during the lipoxygenase-catalyzed reaction. Filtered soybean standard 15-LOX enzyme prepared in Tris-HCl buffer (10 mM, pH: 7.4). The 15-LOX activity was measured at a wavelength of 940 nm with the help of a plate reader brought by BioTek ELX800 (BioTek Instruments, Colmar, France). The negative control was prepared by adding solvent mixture without extract (addition of the same volume of extraction solvent (i.e., methanol)). The inhibition activity was measured as the % inhibition corresponding to this negative control. Nordihydroguaiaretic acid (100 μM) was used as a standard drug inhibitor for 15-LOX enzyme.

### 2.10.3. sPLA2-Inhibition Assay

The inhibition against secretory sPLA2 (phospholipase A2) was investigated through the kit method (Cayman Chem, 10004883; Montluçon, France). Diheptanoyl thio-PC ($C_{22}H_{44}NO_6PS_2$) was taken as a substrate for the enzyme, whereas thiotheramide-PC was considered as a positive control. The reaction was followed using absorbance recorded at 420 nm using a microplate reader (BioTek ELX800, Colmar, France). The negative control was prepared by adding solvent mixture without extract (addition of the same volume of extraction solvent (i.e., methanol)). The inhibition activity was measured as the % inhibition corresponding to this negative control. Thioetheramide-PC (5 μM) was used as a standard sPLA2 enzyme inhibitor.

### 2.11. Statistical Analysis

All the experiments were performed at least in triplicate (at least 3 independent biological replicates). The results were expressed in the form of the means and standard deviations of these replicates. The significant differences between the groups in all the experiments were investigated using ANOVA, followed by 2-tailed multiple *t*-tests with a

Bonferroni correction performed using the XL-STAT 2019 biostatistics software (Addinsoft, Paris, France). All these results were considered significantly different at $p < 0.05$.

## 3. Results and Discussion

### 3.1. Callogenesis

Leaf explants of *S. virginianum* in vitro plantlets grown on MS media supplemented with several PGRs (TDZ, BAP, NAA, and TDZ + NAA) at varying concentrations (0.1, 1, 2.5, 5, and 10 mg/L) were used to assess the callus-induction frequency in relation to each hormonal treatment. All plant growth hormones showed a callogenic capacity when applied to leaf explants after 6–12 days, with a callus-induction frequency ranging from 20% to 100% (Table 1).

**Table 1.** Physiological parameters with FW and DW analyses of callus cultures.

| PGR Concentration (mg/L) | FW (g/L) | DW (g/L) | Callus-Initiation Days | Callogenic Frequency (%) | Callus Color | Callus Texture |
|---|---|---|---|---|---|---|
| 0.1 TDZ | 169.53 ± 3.63 [fgh] | 9.99 ± 0.41 [c] | 7 | 80 % | DG | C |
| 1.0 TDZ | 219.33 ± 5.68 [c] | 10.32 ± 0.44 [c] | 7 | 80 % | DG | C |
| 2.5 TDZ | 246.11 ± 5.72 [a] | 13.03 ± 0.59 [ab] | 6 | 100 % | DG | C |
| 5.0 TDZ | 251.48 ± 5.19 [a] | 13.59 ± 0.63 [a] | 6 | 100 % | DG | C |
| 10.0 TDZ | 244.03 ± 5.88 [a] | 12.72 ± 0.47 [ab] | 6 | 80 % | DG | C |
| 0.1 BAP | 26.12 ± 0.62 [n] | 2.95 ± 0.13 [g] | 10 | 20 % | YG | C |
| 1.0 BAP | 84.01 ± 1.71 [m] | 5.21 ± 0.29 [f] | 9 | 40 % | YG | C |
| 2.5 BAP | 165.26 ± 1.29 [g] | 9.56 ± 0.26 [c] | 9 | 60 % | YG | C |
| 5.0 BAP | 167.33 ± 2.73 [fgh] | 9.88 ± 0.31 [c] | 9 | 80 % | YG | C |
| 10.0 BAP | 172.79 ± 2.21 [f] | 9.99 ± 0.37 [c] | 9 | 60 % | DG | C |
| 0.1 NAA | 109.14 ± 1.88 [k] | 5.78 ± 0.29 [ef] | 10 | 40% | LG | F |
| 1.0 NAA | 187.39 ± 2.37 [e] | 10.00 ± 0.41 [c] | 9 | 80% | LG | F |
| 2.5 NAA | 199.62 ± 2.91 [d] | 10.07 ± 0.37 [c] | 8 | 80% | YG | F |
| 5.0 NAA | 169.48 ± 2.52 [g] | 9.94 ± 0.33 [c] | 10 | 60% | YG | F |
| 10.0 NAA | 93.85 ± 2.11 [i] | 5.80 ± 0.27 [ef] | 12 | 40% | YG | F |
| 0.1 TDZ + 1.0 NAA | 118.99 ± 2.73 [j] | 6.60 ± 0.39 [e] | 8 | 80% | DG | C |
| 1.0 TDZ + 1.0 NAA | 159.48 ± 3.49 [h] | 8.61 ± 0.44 [d] | 8 | 80% | DG | C |
| 2.5 TDZ + 1.0 NAA | 231.40 ± 5.44 [b] | 11.90 ± 0.49 [b] | 9 | 80% | DG | C |
| 5.0 TDZ + 1.0 NAA | 238.90 ± 5.69 [ab] | 12.20 ± 0.41 [b] | 7 | 100% | DG | C |
| 10.0 TDZ + 1.0 NAA | 144.15 ± 2.86 [i] | 7.85 ± 0.28 [d] | 9 | 80% | DG | C |

FW: fresh weight; DW: dry weight, DG: dark green; YG: yellowish green; LG: light green; F: friable; C: compact. Data are expressed as means ± standard deviations of $n = 3$ independent measurements. Different letters represent significant differences between the various extracts ($p < 0.05$).

When compared with the other treatments (BAP, NAA, TDZ + NAA), the TDZ showed a precocial response for callus initiation (6–7 days). Among all the concentrations, TDZ combined with 1 mg/L of NAA, or alone, showed the highest callogenic response, as compared with the rest of the hormones. Moreover, with TDZ alone (2.5 mg/L and 5 mg/L) and in combination (5 mg/L TDZ + 1 mg/L), NAA showed 100% callogenic frequency on the MS media. Morphological variations were also observed in all the calluses, as shown in Figure 1, which indicates that the calli treated with NAA alone were friable, whereas the calli treated with the remaining plant growth regulators were compact. According to a previous report, TDZ is one of the many used phytohormones for the consecutive development of in vitro plant cultures [31]. Because TDZ primarily displays cytokinin-like activity with auxin-like effects, it promotes plant growth and callus formation by stimulating the accumulation of the endogenous auxins in exposed tissues. We can hypothesize that the exogenous presentation of TDZ, either in combination with NAA or alone, influences endogenous growth parameters, thereby enhancing callus induction and growth. A research study conducted by Abbasi et al. observed that TDZ, either with NAA or alone, significantly enhances the callogenesis in the leaf and stem explants of *Isodon*

*rugosus* [32]. Similarly, reported studies have found that either TDZ with NAA or alone greatly enhances the callogenic frequency in cultures of *Artemisia absinthium* L. and *Lepidium sativum* [33,34]. MS media devoid of plant hormone demonstrated no callogenesis, which has already been witnessed for other different plant species, such as *Lepidium sativum* [33].

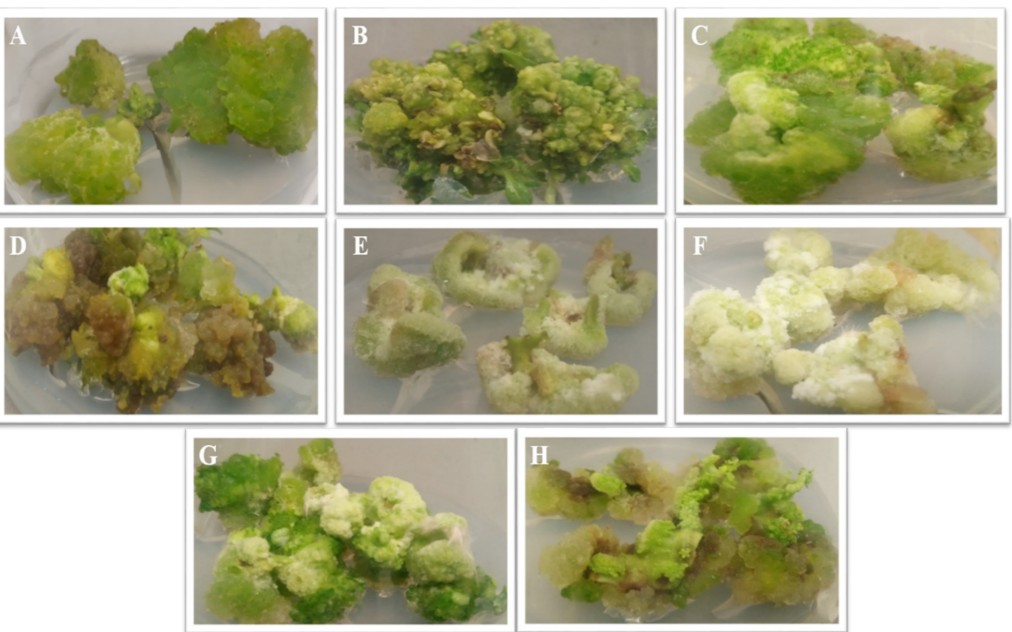

**Figure 1.** Callus culture treatment under various PGR concentrations, callus morphology, and biomass from leaf explant after 30 days of inoculation: (**A**) 5 mg/L TDZ; (**B**) 2.5 mg/L TDZ; (**C**) 5 mg/L TDZ + 1 mg/L NAA; (**D**) 10 mg/L BAP; (**E**) 10 mg/L NAA; (**F**) 2.5 mg/L NAA; (**G**) 2.5 mg/L TDZ + 1 mg/L NAA; (**H**) 5 mg/L BAP.

*3.2. Biomass Accumulation*

Among all the applied plant hormones, TDZ has shown a better response in terms of the biomass accumulation for leaf-derived explants. In the current study, the highest fresh weight (251.48 ± 5.19 g/L) and dry weight (13.59 ± 0.63 g/L) were accumulated in callus grown on media (MS) augmented with 5 mg/L of TDZ, which further notified the cultures supplemented with 2.5 mg/L and 10 mg/L of TDZ, respectively, as shown in Table 1. Similarly, 1 mg/L NAA + 5 mg/L TDZ resulted in an FW of 238.90 ± 5.69 g/L and a DW of 12.20 ± 0.41 g/L, which are the maximums in all the applied NAA + TDZ combinations on the leaf explants. BAP showed a lesser response in terms of the biomass compared with the other PGRs, having an accumulation of biomass in FW of 172.79 ± 2.21 g/L, and in DW of 9.99 ± 0.37 g/L, at the applied concentration of 10 mg/L. For NAA, leaf-derived callus with optimum production (FW: 199.62 ± 2.91 g/L and DW: 10.07 ± 0.37 g/L) was witnessed at a concentration of 2.5 mg/L. Our results follow the study of Ullah et al. [33], where the TDZ showed higher biomass production in the leaf-derived culture of *Lepidium sativum*. Similarly, Abbasi et al. observed that TDZ greatly enhanced the biomass production in the stem-derived callus culture of *Isodon rugosus* [32]. In contrast, Nadeem et al. observed the highest biomass production in the callus culture of *Ocimum basilicum* supplemented with NAA [35].

*3.3. Total Phenolic and Total Flavonoid Contents*

The exogenous presentation of PGRs directly influenced the production of phenolics and flavonoids in the callus cultures of plants, thus enhancing the stress-tolerance potential of the plant [36,37]. In the current study, the TPC and TFC were determined in the leaf-induced callus cultures of *S. virginianum* treated with various plant hormones. Among all the tested PGRs, TDZ showed the highest accumulation of TPC and TFC, followed by BAP. NAA, either alone or in combination with TDZ, showed a lower biosynthesis of phenols and flavonoids. The highest total phenolic accumulation (10.22 ± 1.19 mg/g DW)

was estimated in cultures grown on MS media augmented with 5 mg/L of TDZ. This further resulted in the maximum growth on the media comprising 10 mg/L of TDZ (9.75 ± 0.88 mg/g DW) and 2.5 mg/L of TDZ (9.61 ± 1.03 mg/g DW), as shown in Table 2, whereas a lower accumulation of total phenolics was observed at 0.1 mg/L of NAA (4.50 ± 0.23 mg/g DW). Similarly, the TPP was also higher at 5 mg/L of TDZ (138.90 ± 2.90 mg/L), and lower at 0.1 mg/L of BAP (14.49 ± 0.88 mg/L). As far as the flavonoid biosynthesis is concerned, the highest TFC and TFP accumulations (1.65 ± 0.11 mg/g DW and 21.01 ± 0.61 mg/L, respectively) were shown by 10 mg/L of TDZ, which was further notified by 5 mg/L of TDZ (TFC: 1.48 ± 0.16 mg/L and TFP: 20.16 ± 0.48 mg/L). A lower accumulation of total flavonoid contents (0.47 ± 0.06 mg/g DW) was observed for cultures grown on media containing 10 mg/L of NAA. A total of 0.1 BAP showed lower TFP (2.02 ± 0.31 mg/L). According to previous literature, TDZ enhanced the accumulation of flavonoid and phenolic contents in plants under stress conditions by activating the phenylpropanoid pathway [38,39]. Our findings of the highest flavonoid and phenolic accumulations in cultures treated with TDZ are supported by the findings of Khurshid et al. [40], where we observed the highest accumulation of phenols and flavonoids. Similarly, Ali and Abbasi [41] observed the optimum accumulation of flavonoids and phenolics in the callus culture of *Artemisia absinthium* fortified with TDZ.

**Table 2.** Total phenolic and total flavonoid analyses of *S. virginianum*-treated cultures.

| PGR Concentration (mg/L) | TPC (mg/g DW) | TFC (mg/g DW) | TPP (mg/L) | TFP (mg/L) |
|---|---|---|---|---|
| 0.1 TDZ | 8.98 ± 0.29 [b] | 0.89 ± 0.13 [de] | 89.77 ± 2.41 [de] | 8.89 ± 0.41 [f] |
| 1.0 TDZ | 8.95 ± 0.26 [bc] | 0.94 ± 0.11 [cd] | 92.44 ± 2.22 [d] | 9.80 ± 0.29 [e] |
| 2.5 TDZ | 9.61 ± 0.49 [ab] | 1.16 ± 0.09 [bc] | 125.27 ± 2.81 [b] | 15.18 ± 0.52 [b] |
| 5.0 TDZ | 10.22 ± 0.44 [a] | 1.48 ± 0.16 [ab] | 138.90 ± 2.90 [a] | 20.16 ± 0.48 [a] |
| 10.0 TDZ | 9.75 ± 0.49 [ab] | 1.65 ± 0.11 [a] | 124.13 ± 2.88 [b] | 21.0 ± 0.61 [a] |
| 0.1 BAP | 4.90 ± 0.19 [g] | 0.68 ± 0.09 [e] | 14.49 ± 0.88 [l] | 2.02 ± 0.31 [k] |
| 1.0 BAP | 8.31 ± 0.27 [c] | 0.87 ± 0.06 [d] | 43.34 ± 1.78 [i] | 4.55 ± 0.29 [i] |
| 2.5 BAP | 9.46 ± 0.21 [b] | 1.29 ± 0.09 [b] | 90.45 ± 2.23 [de] | 12.33 ± 1.19 [cd] |
| 5.0 BAP | 8.98 ± 0.28 [b] | 0.74 ± 0.08 [e] | 88.75 ± 2.19 [de] | 7.32 ± 0.31 [g] |
| 10.0 BAP | 8.88 ± 0.35 [bc] | 0.84 ± 0.08 [de] | 88.77 ± 2.20 [de] | 8.43 ± 0.44 [fg] |
| 0.1 NAA | 4.50 ± 0.23 [g] | 0.50 ± 0.08 [f] | 26.0 ± 1.93 [k] | 2.89 ± 0.16 [j] |
| 1.0 NAA | 6.57 ± 0.29 [f] | 0.76 ± 0.12 [de] | 65.78 ± 2.21 [g] | 7.62 ± 0.42 [hg] |
| 2.5 NAA | 7.45 ± 0.27 [de] | 0.87 ± 0.14 [de] | 75.04 ± 1.95 [f] | 8.78 ± 0.37 [f] |
| 5.0 NAA | 8.50 ± 0.48 [bc] | 0.84 ± 0.11 [de] | 84.55 ± 2.20 [e] | 8.38 ± 0.32 [f] |
| 10.0 NAA | 5.94 ± 0.49 [f] | 0.47 ± 0.06 [f] | 34.46 ± 0.98 [j] | 2.72 ± 0.21 [j] |
| 0.1 TDZ + 1.0 NAA | 4.55 ± 0.29 [g] | 0.69 ± 0.09 [e] | 30.06 ± 1.18 [k] | 4.58 ± 0.28 [i] |
| 1.0 TDZ + 1.0 NAA | 6.83 ± 0.31 [ef] | 0.63 ± 0.09 [ef] | 58.83 ± 1.27 [h] | 5.42 ± 0.41 [i] |
| 2.5 TDZ + 1.0 NAA | 7.24 ± 0.34 [e] | 0.84 ± 0.14 [de] | 86.212.21 [e] | 10.08 ± 0.66 [de] |
| 5.0 TDZ + 1.0 NAA | 8.91 ± 0.52 [bc] | 0.95 ± 0.17 [cd] | 108.813.32 [c] | 11.67 ± 0.61 [c] |
| 10.0 TDZ + 1.0 NAA | 8.25 ± 0.41 [cd] | 0.97 ± 0.16 [cd] | 64.78 ± 2.21 [g] | 7.65 ± 0.49 [fgh] |

TPC: total phenolic contents; TFC: total flavonoid contents; TTP: total phenolic production; TFP: total flavonoid production. Data are expressed as means ± standard deviations of $n$ = 3 independent measurements. Different letters represent significant differences between the various extracts ($p < 0.05$).

### 3.4. HPLC Analysis

In a recent study, two coumarins, including scopoletin and esculetin, as well as caffeic acid and one of its derivatives (i.e., methyl caffeate), were quantified through HPLC [8]. Coumarins and caffeic acid derivatives are produced through the phenylpropanoid pathway from *trans*-cinnamic acid [41]. Coumarins are thought to be efficient at producing phytochemicals that play a significant role in the plant defence response against oxidative stress, hormonal stresses, or microbial infections. Herein, the highest accumulation of coumarins was observed in cultures supplemented with TDZ, while the callus treated with BAP showed a lower accumulation of coumarins. The optimum accumulations of the coumarin derivatives esculetin (0.87 ± 0.040 mg/g DW) and scopoletin

(4.34 ± 0.20 mg/g DW) were observed at 10 mg/L of TDZ. While the lowest production of esculetin (0.62 ± 0.032 mg/g DW) and scopoletin (2.32 ± 0.17 mg/g DW) was observed for 0.1 mg/L of BAP, as summarized in Table 3.

**Table 3.** Phytochemical analyses of extracts from treated *S. virginianum* callus cultures.

| PGR Concentration (mg/L) | Caffeic Acid (mg/g DW) | Methyl Caffeate (mg/g DW) | Scopoletin (mg/g DW) | Esculetin (mg/g DW) |
|---|---|---|---|---|
| 0.1 TDZ | 0.46 ± 0.020 [cd] | 14.30 ± 0.66 [def] | 3.72 ± 0.23 [bcd] | 0.72 ± 0.039 [cd] |
| 1.0 TDZ | 0.43 ± 0.017 [d] | 13.07 ± 0.48 [fg] | 3.48 ± 0.21 [cd] | 0.74 ± 0.041 [bc] |
| 2.5 TDZ | 0.52 ± 0.024 [ab] | 16.71 ± 0.61 [b] | 3.32 ± 0.21 [cde] | 0.79 ± 0.044 [ab] |
| 5.0 TDZ | 0.56 ± 0.021 [a] | 18.62 ± 0.60 [a] | 4.07 ± 0.24 [ab] | 0.84 ± 0.043 [ab] |
| 10.0 TDZ | 0.48 ± 0.020 [bc] | 15.33 ± 0.44 [cd] | 4.34 ± 0.20 [a] | 0.87 ± 0.040 [a] |
| 0.1 BAP | 0.45 ± 0.021 [cd] | 11.70 ± 0.40 [h] | 2.32 ± 0.17 [f] | 0.62 ± 0.032 [e] |
| 1.0 BAP | 0.54 ± 0.024 [a] | 15.65 ± 0.43 [bc] | 2.33 ± 0.19 [f] | 0.63 ± 0.038 [de] |
| 2.5 BAP | 0.55 ± 0.026 [a] | 18.19 ± 0.61 [a] | 2.98 ± 0.21 [e] | 0.71 ± 0.033 [c] |
| 5.0 BAP | 0.49 ± 0.018 [bc] | 14.95 ± 0.63 [de] | 2.99 ± 0.22 [e] | 0.70 ± 0.039 [cd] |
| 10.0 BAP | 0.48 ± 0.018 [bc] | 15.05 ± 0.66 [de] | 3.57 ± 0.20 [bcd] | 0.77 ± 0.036 [bc] |
| 0.1 NAA | 0.37 ± 0.020 [ef] | 9.42 ± 0.39 [ij] | 3.20 ± 0.20 [cde] | 0.73 ± 0.030 [c] |
| 1.0 NAA | 0.43 ± 0.018 [de] | 12.56 ± 0.37 [g] | 3.66 ± 0.21 [bcd] | 0.79 ± 0.032 [ab] |
| 2.5 NAA | 0.45 ± 0.019 [cd] | 12.77 ± 0.34 [g] | 2.86 ± 0.20 [e] | 0.68 ± 0.031 [cde] |
| 5.0 NAA | 0.47 ± 0.026 [cd] | 14.39 ± 0.38 [de] | 3.25 ± 0.21 [de] | 0.73 ± 0.040 [bc] |
| 10.0 NAA | 0.37 ± 0.024 [ef] | 9.70 ± 0.41 [i] | 3.33 ± 0.27 [de] | 0.75 ± 0.038 [bc] |
| 0.1 TDZ + 1.0 NAA | 0.39 ± 0.024 [e] | 10.41 ± 0.39 [hi] | 2.99 ± 0.18 [e] | 0.70 ± 0.029 [cd] |
| 1.0 TDZ + 1.0 NAA | 0.33 ± 0.021 [f] | 8.71 ± 0.33 [j] | 3.39 ± 0.29 [cde] | 0.73 ± 0.033 [c] |
| 2.5 TDZ + 1.0 NAA | 0.40 ± 0.024 [e] | 11.48 ± 0.37 [h] | 3.40 ± 0.29 [cde] | 0.74 ± 0.032 [bc] |
| 5.0 TDZ + 1.0 NAA | 0.40 ± 0.021 [e] | 12.04 ± 0.51 [gh] | 3.94 ± 0.22 [bc] | 0.81 ± 0.041 [ab] |
| 10.0 TDZ + 1.0 NAA | 0.46 ± 0.020 [d] | 14.10 ± 0.50 [ef] | 3.77 ± 0.22 [bc] | 0.80 ± 0.040 [ab] |

Data are expressed as means ± standard deviations of $n = 3$ independent measurements. Different letters represent significant differences between the various extracts ($p < 0.05$).

Caffeic acids and their derivatives are thought to be involved in lignin synthesis, the regulation of water flux, phototropism, turgor pressure, and the expansion and growth of plant cells [42]. In addition, it also protects the plant from certain biotic and abiotic stress conditions, including microbial infections, salinity stress, drought, extreme temperature, UV light, and metal stress [43]. In the current study, two caffeic acid derivatives (i.e., methyl caffeate and caffeic acid itself), in callus cultures of *S. virginianum* treated with different types of plant hormones, were quantified through HPLC. The highest accumulation of caffeic acid (0.56 ± 0.021 mg/g DW) was observed for cultures fortified with 5 mg/L of TDZ, as well as for 1 mg/L of BAP (0.54 ± 0.024 mg/g DW) and 2.5 mg/L of BAP (0.55 ± 0.026 mg/g DW). The lowest production of caffeic acid was observed for 1 mg/L TDZ + 1 mg/L NAA (0.33 ± 0.021 mg/g DW). Likewise, the highest biosynthesis of methyl caffeate was noted for cultures supplied with 5 mg/L of TDZ (18.62 ± 0.60 mg/g DW) and 2.5 mg/L of BAP (18.19 ± 0.61 mg/g DW). The phenylpropanoid pathway is a source of very distinct (poly) phenolic compounds that are biosynthesized through the activity of PAL enzymes, as described in [42].

### 3.5. Antioxidant Evaluation

The production of ROS and other free radicals causes metabolic pathways to revert due to environmental stress, which is responsible for damaging membranes, plant cells, DNA, lipids, and proteins, directly or indirectly [43]. Oxidative stress causes plants to synthesize various antioxidant phytochemicals, and mainly flavonoids, terpenoids, and phenolics, which are considered antioxidants that quench these free radicals [44,45]. In this study, *S. virginianum* callus extract was used to determine three antioxidant assays: a ferric reducing antioxidant potential (FRAP) assay, based on electron transfer (ET)-based antioxidant activity; a 2,2-azinobis-3-ethylbenzthiazoline-6-sulphonic acid assay (ABTS); a 2,2-Diphenyl-1-picrylhydrazyl) (DPPH) assay involving a hydrogen atom transfer (HAT)-

based antioxidant mechanism. The DPPH value was expressed as the percentage of free-radical-scavenging activity (RSA), while the FRAP and ABTS assays were evaluated according to the Trolox C equivalent antioxidant capacity (μM TEAC). The maximum FRAP and ABTS antioxidant activities were obtained for the extract from *S. virginianum* callus treated with 5 mg/L of TDZ (654 ± 5.39 μM TEAC and 402.5 ± 5.16 μM TEAC, respectively), followed by the extract from *S. virginianum* callus treated with 2.5 mg/L of BAP (FRAP: 653.3 ± 5.62 μM TEAC, and ABTS: 383.2 ± 4.48 μM TEAC). Among all the extracts, the lowest FRAP and ABTS antioxidant potentials were shown for the extract obtained from *S. virginianum* callus treated with 0.1 mg/L TDZ + 1 mg/L NAA (FRAP: 312.2 ± 4.36 μM TEAC, and ABTS: 212.85 ± 3.81 mM TEAC). Similarly, the highest DPPH antioxidant values were measured for the extract obtained from *S. virginianum* callus treated with 10 mg/L of TDZ (94.6 ± 2.29% RSA), 2.5 mg/L of BAP (92.6 ± 3.10% RSA), and 5 mg/L of TDZ (92 ± 3.49% RSA), whereas the lowest DPPH antioxidant activity was recorded for the extract from the *S. virginianum* callus treated with 10 mg/L of NAA (71.4% RSA) (Table 4). Here, for callus cultures of *S. virginianum* treated with various hormones, we found a positive correlation between the antioxidant capacity and phytochemical production. This observation is in agreement with the results of Nazir et al. [46], who reported the highest antioxidant capacity for extract treated with 5 mg/L of TDZ, when compared with other hormonal treatments, in *Ocimum basilicum* callus cultures. The antioxidant activities of plant extracts are considered complex mechanisms that are controlled by multiple factors that are influenced by the compositions of the extracts. Thus, it was considered important to investigate the antioxidant capacity of the callus extracts involved in more than one form of the free-radical-scavenging test [47].

**Table 4.** Potential in vitro antioxidant (DPPH, FRAP, and ABTS) activities of the extracts of callus cultures of *S. virginianum* treated with different plant growth regulators.

| PGR Concentration (mg/L) | ABTS (μM TEAC) | FRAP (μM TEAC) | DPPH (% RSA) |
|---|---|---|---|
| 0.1 TDZ | 212.85 ± 3.81 [i] | 312.2 ± 4.36 [j] | 88.1 ± 4.29 [bc] |
| 1.0 TDZ | 187.34 ± 4.18 [k] | 267.1 ± 4.24 [k] | 76.3 ± 3.72 [de] |
| 2.5 TDZ | 154.82 ± 3.49 [l] | 461.7 ± 4.98 [e] | 89.7 ± 3.84 [abc] |
| 5.0 TDZ | 402.51 ± 5.16 [a] | 654.6 ± 5.39 [a] | 92 ± 3.49 [ab] |
| 10.0 TDZ | 370.49 ± 5.62 [c] | 510.9 ± 6.16 [b] | 94.6 ± 2.29 [ab] |
| 0.1 BAP | 187.38 ± 4.38 [j] | 378.1 ± 4.62 [gh] | 72.7 ± 3.94 [e] |
| 1.0 BAP | 348.61 ± 6.63 [d] | 469.1 ± 4.29 [d] | 79.3 ± 4.11 [cde] |
| 2.5 BAP | 383.2 ± 4.48 [b] | 653.3 ± 5.62 [a] | 92.6 ± 3.10 [ab] |
| 5.0 BAP | 311.7 ± 4.61 [e] | 489.6 ± 4.70 [c] | 88.5 ± 4.84 [ab] |
| 10.0 BAP | 339.3 ± 4.44 [d] | 519.2 ± 5.81 [b] | 90.2 ± 4.17 [ab] |
| 0.1 NAA | 254.7 ± 3.69 [h] | 378.4 ± 4.88 [g] | 78.3 ± 3.22 [cde] |
| 1.0 NAA | 298.9 ± 3.67 [f] | 454.7 ± 6.17 [de] | 82.9 ± 3.96 [cd] |
| 2.5 NAA | 310.3 ± 3.49 [e] | 411.9 ± 6.29 [f] | 87.5 ± 3.72 [abc] |
| 5.0 NAA | 317.1 ± 4.49 [e] | 468.9 ± 5.16 [d] | 91.2 ± 4.19 [ab] |
| 10.0 NAA | 278.9 ± 4.16 [g] | 382.1 ± 5.44 [g] | 87.7 ± 4.40 [abc] |
| 0.1 TDZ + 1.0 NAA | 293.9 ± 4.38 [f] | 349.7 ± 5.93 [i] | 89.2 ± 4.27 [abc] |
| 1.0 TDZ + 1.0 NAA | 311.4 ± 5.81 [e] | 367.3 ± 5.37 [hi] | 91.9 ± 4.51 [abc] |
| 2.5 TDZ + 1.0 NAA | 303.6 ± 5.89 [ef] | 391.2 ± 4.87 [g] | 91.1 ± 3.93 [abc] |
| 5.0 TDZ + 1.0 NAA | 307.5 ± 4.74 [ef] | 418.4 ± 4.95 [f] | 86.6 ± 3.72 [bcd] |
| 10.0 TDZ + 1.0 NAA | 319.9 ± 5.73 [e] | 449.9 ± 5.81 [e] | 90.6 ± 4.61 [abc] |

ABTS and FRAP are expressed in μM Trolox C equivalent antioxidant capacity (TEAC). DPPH is expressed as % of radical scavenging activity (RSA). Trolox was used as standard for ABTS and FRAP activity. Ascorbic acid was used as standard for DPPH assay. Data are expressed as means ± standard deviations of $n = 3$ independent measurements. Different letters represent significant differences between the various extracts ($p < 0.05$).

## 3.6. Antiaging Evaluation

The traditional use of *S. virginianum* extracts for the skin appearance has been reported [48]. Therefore, the skin antiaging potency of *S. virginianum* callus extracts was



evaluated for inhibiting, in vitro, tyrosinase, elastase, collagenase, hyaluronidase, and AGEs (vesperlysine-like and pentosidine-like AGEs). The results expressed as inhibition percentages (%) are given in Table 5. The extract obtained from the callus treated with 5 mg/L of TDZ showed the highest inhibition against vesperlysine-like AGEs ($51.55 \pm 2.67\%$) and the formation and inhibition of tyrosinase ($32.87 \pm 2.04\%$) and collagenase ($49.52 \pm 2.69\%$) enzymes, while the extract obtained from the callus treated with 2.5 mg/L of TDZ showed the maximum inhibition ($59.75 \pm 3.15$) against pentosidine-like AGE formation. It should be noted that no inhibition was observed for elastase (less than 10% inhibition) and hyaluronidase (less than 5% inhibition), which indicates that the current extracts may have skin antiaging potential primarily through the inhibitions of AGE formation, as well as tyrosinase and collagenase actions. According to our assessment, this is the first study performed to analyze the antiaging capacity of *S. virginianum* callus cultures treated with TDZ, BAP, and NAA. Apart from this, there is a plethora of literature available that assesses the antiaging activities of callus cultures that have been enhanced with the application of different elicitors [49]. In agreement with our observations, caffeic acid, methyl caffeate, and its derivatives were evaluated as potent inhibitors of tyrosinase [50], whereas scopoletin and esculetin showed remarkable inhibition toward collagenase [51].

**Table 5.** Potential in vitro skin antiaging activities (expressed in % of inhibition) of the extracts of callus cultures of *S. virginianum* treated with different plant growth regulators.

| PGR Concentration (mg/L) | Vesperlysine AGEs | Pentosidine AGEs | Tyrosinase | Elastase | Hyaluronidase | Collagenase |
|---|---|---|---|---|---|---|
| 0.1 TDZ | $42.87 \pm 2.21$ [cde] | $50.78 \pm 3.11$ [c] | $25.48 \pm 1.84$ [cd] | $1.35 \pm 0.18$ [i] | $2.91 \pm 0.19$ [bc] | $46.40 \pm 2.11$ [ab] |
| 1.0 TDZ | $40.49 \pm 1.86$ [def] | $52.77 \pm 3.24$ [abc] | $25.65 \pm 1.87$ [cd] | $5.36 \pm 0.66$ [bcd] | $2.87 \pm 0.21$ [bc] | $45.65 \pm 2.20$ [abc] |
| 2.5 TDZ | $47.19 \pm 1.89$ [abc] | $59.75 \pm 3.15$ [a] | $30.71 \pm 2.01$ [ab] | $4.78 \pm 0.66$ [cd] | $2.62 \pm 0.21$ [cde] | $41.80 \pm 2.29$ [bcd] |
| 5.0 TDZ | $51.55 \pm 2.67$ [a] | $57.12 \pm 3.10$ [ab] | $32.87 \pm 2.04$ [a] | $4.65 \pm 0.43$ [cd] | $3.14 \pm 0.31$ [ab] | $49.52 \pm 2.69$ [a] |
| 10.0 TDZ | $45.07 \pm 1.96$ [c] | $53.84 \pm 3.19$ [abc] | $28.42 \pm 1.67$ [bc] | $8.52 \pm 0.44$ [a] | $3.38 \pm 0.19$ [a] | $43.68 \pm 2.20$ [abc] |
| 0.1 BAP | $33.47 \pm 1.62$ [h] | $36.12 \pm 2.89$ [d] | $17.15 \pm 1.19$ [g] | $1.78 \pm 0.09$ [h] | $1.67 \pm 0.16$ [f] | $22.53 \pm 2.21$ [f] |
| 1.0 BAP | $40.83 \pm 2.09$ [e] | $55.67 \pm 2.96$ [abc] | $26.54 \pm 1.64$ [bcd] | $4.56 \pm 0.23$ [d] | $1.65 \pm 0.16$ [f] | $26.35 \pm 1.82$ [f] |
| 2.5 BAP | $49.11 \pm 2.20$ [ab] | $52.36 \pm 2.91$ [bc] | $32.28 \pm 1.51$ [a] | $6.12 \pm 0.34$ [b] | $2.24 \pm 0.19$ [de] | $37.76 \pm 1.89$ [de] |
| 5.0 BAP | $41.34 \pm 1.94$ [de] | $53.03 \pm 3.11$ [abc] | $24.12 \pm 1.50$ [de] | $2.56 \pm 0.23$ [fg] | $2.27 \pm 0.20$ [de] | $36.41 \pm 1.86$ [e] |
| 10.0 BAP | $43.46 \pm 1.79$ [cde] | $57.88 \pm 3.20$ [ab] | $26.47 \pm 1.29$ [cd] | $2.89 \pm 0.20$ [ef] | $2.81 \pm 0.19$ [bc] | $44.74 \pm 2.65$ [abc] |
| 0.1 NAA | $26.47 \pm 1.54$ [i] | $56.60 \pm 3.19$ [ab] | $18.40 \pm 1.24$ [fg] | $2.85 \pm 0.21$ [ef] | $2.40 \pm 0.20$ [de] | $37.26 \pm 2.10$ [de] |
| 1.0 NAA | $36.78 \pm 1.39$ [f] | $57.26 \pm 3.44$ [ab] | $21.28 \pm 1.20$ [ef] | $2.74 \pm 0.24$ [fg] | $2.79 \pm 0.21$ [bc] | $41.38 \pm 2.17$ [bcd] |
| 2.5 NAA | $33.94 \pm 1.30$ [h] | $49.85 \pm 2.79$ [c] | $23.85 \pm 1.17$ [de] | $2.95 \pm 0.20$ [ef] | $2.16 \pm 0.21$ [e] | $36.39 \pm 2.20$ [e] |
| 5.0 NAA | $42.80 \pm 2.10$ [cde] | $59.64 \pm 3.20$ [ab] | $24.87 \pm 1.19$ [de] | $1.45 \pm 0.19$ [hi] | $2.51 \pm 0.20$ [cde] | $38.04 \pm 2.12$ [de] |
| 10.0 NAA | $39.63 \pm 1.87$ [def] | $50.63 \pm 3.19$ [bc] | $16.17 \pm 1.04$ [g] | $5.63 \pm 0.44$ [bc] | $2.54 \pm 0.21$ [cde] | $37.42 \pm 2.19$ [de] |
| 0.1 TDZ + 1.0 NAA | $41.32 \pm 2.11$ [de] | $54.33 \pm 3.22$ [abc] | $18.12 \pm 1.0$ [fg] | $0.63 \pm 0.07$ [j] | $2.32 \pm 0.20$ [de] | $36.95 \pm 2.30$ [de] |
| 1.0 TDZ + 1.0 NAA | $39.82 \pm 1.93$ [def] | $57.34 \pm 3.24$ [ab] | $17.24 \pm 1.11$ [g] | $1.36 \pm 0.09$ [hi] | $2.79 \pm 0.26$ [bcd] | $39.33 \pm 2.31$ [de] |
| 2.5 TDZ + 1.0 NAA | $45.61 \pm 1.94$ [bcd] | $55.39 \pm 3.21$ [abc] | $23.65 \pm 2.0$ [de] | $2.45 \pm 0.11$ [g] | $2.73 \pm 0.23$ [bcd] | $37.42 \pm 2.30$ [de] |
| 5.0 TDZ + 1.0 NAA | $42.38 \pm 2.11$ [d] | $58.68 \pm 3.20$ [ab] | $23.33 \pm 2.03$ [de] | $4.12 \pm 0.23$ [d] | $3.20 \pm 0.26$ [ab] | $40.93 \pm 2.44$ [cde] |
| 10.0 TDZ + 1.0 NAA | $45.58 \pm 2.21$ [bc] | $52.19 \pm 3.10$ [abc] | $24.30 \pm 2.04$ [de] | $3.32 \pm 0.21$ [e] | $2.96 \pm 0.23$ [ab] | $37.21 \pm 2.21$ [de] |

Aminoguanidine (150 µM) was used as the standard inhibitor of AGE formation (inhibitions of $27.3 \pm 3.9\%$ and $32.4 \pm 4.2\%$ of vesperlysine-like and pentosidine-like AGE formation, respectively). Phenantroline (100 µM) was used as the specific inhibitor of collagenase (inhibition of $33.6 \pm 2.2\%$). Kojic acid (10 µM) was used as the specific inhibitor of tyrosinase (inhibition of $51.2 \pm 0.9\%$). Oleanolic acid (10 µM) was used as the specific inhibitor of elastase and hyaluronidase (inhibitions of $47.8 \pm 1.4\%$ and $33.5 \pm 2.8\%$ of elastase and hyaluronidase, respectively). Data are expressed as means $\pm$ standard deviations of $n = 3$ independent measurements. Different letters represent significant differences between the various extracts ($p < 0.05$).

### 3.7. Anti-Inflammatory Evaluation

Inflammation is a natural healing system in reaction to harmful stimuli, pathogens, and irritants. Excessive inflammation can be damaging, and especially to the appearance of the skin. Cyclooxygenases (COX-1 and COX-2) are critical players in inflammation that have been demonstrated to be suppressed by several medicinal plant extracts, which has demonstrated the anti-inflammatory capabilities of these extracts [52,53]. The tradi-

tional use of S. virginianum as an anti-inflammatory agent has been proposed in previous reports [54]. Phenylpropanoids, such as caffeic acid and methyl caffeate produced in S. virginianum callus cultures, have been shown to have anti-inflammatory activity by inactivating the sPLA2, COX-1, COX-2, and 15-LOX enzymes [30], which constitutively decrease the prostanoid and leukotriene concentrations [55]. Furthermore, the anti-inflammatory potential of the present S. virginianum callus culture extracts was estimated using COX-2, COX-1, sPLA2, and 15-LOX assays (Table 6). Despite the absence of a purification step, substantial intermediate inhibitory potential was observed for the S. virginianum callus culture extracts when compared to the measured values for standard inhibitor drugs (thioetheramide-PC for sPLA2, nordihydroguaiaretic acid for 15-LOX, and ibuprofen for COX-1 and COX-2). Here, among all the extracts, the greatest anti-inflammatory action was found for: the extract obtained from the S. virginianum callus culture treated with 0.1 mg/L of TDZ (11.3 ± 1.02%) against sPLA2; the extract obtained from the S. virginianum callus culture treated with 5 mg/L of TDZ (38.5 ± 2.29%) against 15-LOX; the extract obtained from the S. virginianum callus culture treated with 10 mg/L of TDZ (38.06 ± 2.49%) against COX-1; the extract obtained from the S. virginianum callus culture treated with 2.5 mg/L of TDZ (15.5 ± 0.71%) against COX-2.

**Table 6.** Potential in vitro anti-inflammatory activities (expressed in % of inhibition) of the extracts of callus cultures of S. virginianum treated with different plant growth regulators.

| PGR Concentrations (mg/L) | COX-1 | COX-2 | 15-LOX | sPLA2 |
|---|---|---|---|---|
| 0.1 TDZ | 34.52 ± 1.93 [abc] | 11.36 ± 0.66 [bc] | 34.63 ± 1.80 [ab] | 11.34 ± 1.02 [ab] |
| 1.0 TDZ | 30.28 ± 1.20 [d] | 10.25 ± 0.61 [cd] | 32.78 ± 1.67 [b] | 7.78 ± 0.38 [d] |
| 2.5 TDZ | 27.56 ± 1.19 [e] | 15.56 ± 0.71 [a] | 29.12 ± 1.09 [cd] | 7.54 ± 0.33 [d] |
| 5.0 TDZ | 33.12 ± 1.66 [bc] | 14.90 ± 0.72 [a] | 38.53 ± 2.29 [a] | 6.89 ± 0.41 [e] |
| 10.0 TDZ | 38.06 ± 2.49 [a] | 12.36 ± 0.40 [b] | 35.87 ± 2.02 [ab] | 6.41 ± 0.30 [e] |
| 0.1 BAP | 15.63 ± 1.38 [h] | 10.25 ± 0.33 [cd] | 18.48 ± 1.62 [f] | 10.36 ± 0.81 [abc] |
| 1.0 BAP | 17.25 ± 1.41 [h] | 8.50 ± 0.39 [ef] | 17.28 ± 1.72 [f] | 8.41 ± 0.64 [cd] |
| 2.5 BAP | 25.63 ± 1.27 [efg] | 7.41 ± 0.31 [f] | 26.92 ± 1.61 [de] | 11.10 ± 1.11 [ab] |
| 5.0 BAP | 23.14 ± 1.21 [g] | 9.63 ± 0.41 [d] | 25.47 ± 1.86 [de] | 5.89 ± 1.01 [e] |
| 10.0 BAP | 31.20 ± 2.03 [bcd] | 9.63 ± 0.44 [cd] | 31.16 ± 2.11 [bc] | 8.63 ± 0.71 [cd] |
| 0.1 NAA | 25.31 ± 1.81 [ef] | 12.54 ± 0.51 [b] | 26.63 ± 2.03 [d] | 6.13 ± 0.44 [e] |
| 1.0 NAA | 30.36 ± 1.96 [cde] | 14.20 ± 0.62 [a] | 32.86 ± 1.92 [b] | 7.45 ± 0.49 [de] |
| 2.5 NAA | 22.15 ± 1.05 [g] | 12.36 ± 0.49 [b] | 23.65 ± 1.49 [e] | 5.85 ± 0.32 [e] |
| 5.0 NAA | 26.85 ± 1.23 [ef] | 14.36 ± 0.51 [a] | 27.87 ± 1.67 [cde] | 10 ± 0.40 [b] |
| 10.0 NAA | 26.95 ± 1.06 [e] | 14.45 ± 0.62 [a] | 28.13 ± 1.68 [cde] | 11.25 ± 0.61 [a] |
| 0.1 TDZ + 1.0 NAA | 24.15 ± 1.11 [fg] | 11.74 ± 0.42 [b] | 25.73 ± 1.37 [e] | 7.63 ± 0.39 [d] |
| 1.0 TDZ + 1.0 NAA | 30.29 ± 2.08 [cde] | 8.52 ± 0.32 [ef] | 32.92 ± 2.02 [b] | 8.93 ± 0.46 [c] |
| 2.5 TDZ + 1.0 NAA | 29.17 ± 2.01 [cde] | 9.65 ± 0.39 [d] | 35.56 ± 2.21 [ab] | 10.14 ± 0.60 [ab] |
| 5.0 TDZ + 1.0 NAA | 35.70 ± 2.06 [ab] | 9.34 ± 0.31 [de] | 35.42 ± 2.08 [ab] | 8.45 ± 0.48 [cd] |
| 10.0 TDZ + 1.0 NAA | 32.23 ± 2.11 [abcd] | 10.64 ± 0.44 [cd] | 33.87 ± 2.04 [ab] | 6.76 ± 0.40 [e] |

Anti-inflammatory activity of cultures of S. virginianum grown on various concentrations and combinations of PGRs. Ibuprofen (10 μM) was used as standard inhibitor of COX-1 and COX-2 enzymes (inhibitions of 31.4 ± 0.8% and 29.8 ± 1.2% of COX-1 and COX-2, respectively). Thioetheramide-PC (5 μM) was used as standard inhibitor of sPLA2 enzyme (inhibition of 43.7 ± 0.8%). Nordihydroguaiaretic acid (100 μM) was used as standard inhibitor of 15-LOX enzyme (inhibition of 30.6 ± 0.7%). Data are expressed as means ± standard deviations of $n = 3$ independent measurements. Different letters represent significant differences between the various extracts ($p < 0.05$).

### 3.8. Correlation Analysis

To identify possible connections between the phytochemicals and biological activities of the extracts from S. virginianum callus cultures, Pearson correlation coefficients were calculated (Figure 2). The PCC analysis revealed the strongest correlations for caffeic acid and its derivative methyl caffeate with the FRAP (ET-based mechanism) antioxidant activity and skin antiaging action (tyrosinase and elastase), whereas scopoletin and esculetin were

mostly connected to the anti-inflammatory action (COX-1, 15-LOX, as well as COX-2 for scopoletin) and DPPH (HAT-based mechanism) antioxidant assay. The inhibition of AGEs was correlated with the presence of caffeic acid and esculetin for the vesperlysine-like AGEs, and scopoletin and esculetin for the pentosidine-like AGEs. Altogether, these correlations indicated the complex mechanisms that result from the complex nature of the plant extract, as already observed for other plant species [56–58]. In the case of S. virginianum callus cultures, the present correlation analysis offers future directions for further research that aims at elucidating the biological activities of S. virginianum extracts/purified compounds.

| | TPC | TFC | Caffeic acid | Methyl caffeate | Scopoletin | Esculetin |
|---|---|---|---|---|---|---|
| ABTS | 0.250 | 0.086 | 0.295 | 0.333 | 0.240 | 0.273 |
| FRAP | **0.598** ** | 0.425 | **0.727** *** | **0.754** *** | 0.159 | 0.324 |
| DPPH | 0.131 | 0.180 | 0.160 | 0.317 | **0.525** * | **0.540** * |
| Vesperlysine-AGEs | **0.483** * | 0.361 | **0.564** ** | 0.694 | 0.434 | **0.474** * |
| Pentosidine-AGEs | 0.018 | 0.117 | 0.026 | 0.161 | **0.457** * | **0.524** * |
| Tyrosinase | **0.607** *** | **0.496** * | **0.859** *** | **0.942** *** | 0.309 | 0.380 |
| Elastase | 0.411 | 0.092 | 0.389 | **0.445** * | 0.343 | **0.474** * |
| Hyaluronidase | -0.024 | -0.079 | -0.171 | 0.095 | **0.987** *** | **0.917** *** |
| Collagenase | -0.060 | 0.029 | 0.066 | 0.278 | **0.840** *** | **0.762** *** |
| COX-1 | -0.047 | -0.067 | -0.096 | 0.129 | **0.977** *** | **0.882** *** |
| COX-2 | -0.239 | -0.307 | 0.042 | 0.057 | **0.301** * | 0.410 |
| 15-LOX | -0.061 | -0.113 | -0.130 | 0.103 | **0.942** *** | **0.852** *** |
| sPLA2 | 0.285 | 0.411 | 0.115 | 0.098 | -0.248 | -0.292 |

**Figure 2.** Pearson correlation coefficients (PCCs) linking the phytochemicals and biological activities of extracts from S. virginianum callus cultures. Colors indicate the strengths of the PCC values, with low PCC values in blue and high PCC values in red. Only statistically significant PCC values are in bold. *** significant $p < 0.001$; ** significant $p < 0.01$; * significant $p < 0.05$.

## 4. Conclusions

Plant hormones were used alone or in combination with different doses in the current investigation to analyze their effects on the biomass accumulation of *S. virginianum* callus cultures, as well as the phytochemical compositions and biological activities of the corresponding extracts. Three different types of hormones (i.e., TDZ, NAA, BAP), alone and in a combination of TDZ + NAA, were employed. TDZ was observed to be more effective in terms of callus induction, biomass accumulation, phytochemical production, and the resulting biological activities, as compared with BAP, NAA, and TDZ + NAA. Markedly, among the TDZ concentrations studied, 5 mg/L of TDZ resulted in increased total phenolic, methyl caffeate, and caffeic acid contents. The extracts of *S. virginianum* callus cultures treated with 5 mg/L of TDZ were shown to have the highest antioxidant (FRAP and ABTS assays), anti-inflammatory (15-LOX inhibition), and antiaging (collagenase, tyrosinase, and vesperlysine-like AGE inhibition) potentials. The application of TDZ at a concentration of 10 mg/L also showed the capacity to improve the accumulation of both the total flavonoid and coumarin (scopoletin and esculetin) contents. In the future, this research will aid in the large-scale synthesis of essential phytochemicals in *S. virginianum* bioreactors that utilize various culture techniques, such as tissue, organ, and cell culture, for possible industrial production for future cosmetic applications.

**Author Contributions:** Conceptualization, H.U., H.J., S.D., L.G., D.T., C.H. and B.H.A.; methodology, H.U., G.Z., H.J., S.D., L.G. and D.T.; software, M.K., B.H.A.; validation, G.Z., H.J. and B.H.A.; formal analysis, M.K.; investigation, H.U.; data curation, H.U.; writing—original draft preparation, H.U. and H.J.; writing—review and editing, H.U. and H.J.; supervision, C.H. and B.H.A. All authors have read and agreed to the published version of the manuscript.

**Funding:** This research was supported by Cosmetosciences, a global training and research program that is dedicated to the cosmetics industry. Located in the heart of Cosmetic Valley, this program, led by the University of Orléans, is funded by the Région Centre-Val de Loire.

**Institutional Review Board Statement:** Not applicable.

**Informed Consent Statement:** Not applicable.

**Data Availability Statement:** All the data supporting the findings of this study are included in this article.

**Acknowledgments:** We are thankful to the Department of Biotechnology, Quaid-i-Azam University, Pakistan, for their support. B.H.A., C.H. and D.T. acknowledge the research fellowships of Le Studium-Institute for Advanced Studies, Loire Valley, Orléans, France. S.D. acknowledges the research fellowship of Région Centre-Val de Loire.

**Conflicts of Interest:** The authors declare no conflict of interests.

## Abbreviations

| | |
|---|---|
| NAA | Naphthalene acetic acid |
| TDZ | Thidiazuron |
| BAP | Benzylaminopurine |
| MS | Murashige and Skoog |
| HPLC | High-performance liquid chromatography |
| DPPH | 2,2-Diphenyl-1-picrylhydrazyl |
| ABTS | Azinobis-3-Ethylbenzthiazoline-6-Sulphonic acid |
| FRAP | Ferric Reducing Antioxidant Potential |
| TPC | Total phenolic content |
| TPP | Total phenolic production |
| TFC | Total flavonoid content |
| TFP | Total flavonoid production |
| PGRs | Plant growth regulators |
| FW | Fresh weight |
| DW | Dry weight |
| PAL | Phenylalanine ammonia-lyase |
| AGE | Advanced glycation end products |
| sPLA2 | Phospholipase A2 |
| COX-1 | Cyclooxygenase-1 |
| COX-2 | Cyclooxygenase-2 |
| 15-LOX | Lipoxygenase |
| ROS | Reactive oxygen species |
| DMSO | Dimethyl sulfoxide |
| OD | Optical density |

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
