# Peer review of "Comparative Analysis of Various Plant-Growth-Regulator Treatments on Biomass Accumulation, Bioactive Phytochemical Production, and Biological Activity of Solanum virginianum L. Callus Culture Extracts"

_cosmetics, doi:10.3390/cosmetics9040071_

Round 1

Reviewer 1 Report

The manuscript “Optimization of Solanum xanthocarpum Cultures…” shows an interesting work and could be considered for publication in Cosmetics after moderate revision as follows:

1.- English should be revised. In general, spelling is OK, but several mistakes have been detected, such as: line 135 productions -> production; line 140 (AlCl3) -> AlCl3; line 156 combine -> combined; line 165 by (Sigma -> by Sigma; and many others (mashroom instead of mushroom; flavanoid instead of flavonoid, etc.). In particular, please, revise the suitable use of commas and brackets.

2.- Abbreviations must be defined when used the first time: MS, TDZ, BAP, NAA, FW, DW, OD… It is necessary to reach section 3.5 for an appropriate description of some of them.

3.- Section 2.1. Please, describe in detail all the “high-quality salts, chemicals, and solutions used”.

4.- Line 152. HPLC with “standard grade” solvents? In HPLC, highest purity solvents are required.

5.- Line 156 is confusing. Solvent “B” is formic acid or ethanol?

6.- Line 168: apart of calibration graph and R2 coefficient, please, provide detection and quantitation limits as well as the linear range.

7.- Lines 193-194. Please, revise the full sentence.

8.- Line 197: form or “from”?

9.- Line 216: Please, change “at 330 nm excitation wavelength and 440 emission wavelengths.” to “at 330 nm (excitation) and 440 (emission) wavelengths.”

10.- Lines 243 & 251. What do you mean exactly with “lacking sample extract”?

11.- Line 155. Please, revise the sentence. What is the meaning of “treated different”?

12.- Lines 317-319. Is this an internal commentary of a pre-final version of the manuscript? If so, please delete it.

13.- Line 329 and others: Please, make well use of significant figures (13.58889±0.63) -> (13.59±0.63). This revision must be made throughout the manuscript.

14.- The meaning of *, ** and *** is not provided in table 1

15.- Line 405: between mean and SD a “±” symbol must be provided.

16.- Lines 442-449 should be moved to introduction or deleted, since the information is not related with results. The same can be said for lines 468-483.

17.- Conclusions should be condensed. Now they are a mere summary of the paper.

Author Response

The manuscript “Optimization of Solanum xanthocarpum Cultures…” shows an interesting work and could be considered for publication in Cosmetics after moderate revision as follows:

1.- English should be revised. In general, spelling is OK, but several mistakes have been detected, such as: line 135 productions -> production; line 140 (AlCl3) -> AlCl3; line 156 combine -> combined; line 165 by (Sigma -> by Sigma; and many others (mashroom instead of mushroom; flavanoid instead of flavonoid, etc.). In particular, please, revise the suitable use of commas and brackets.

Response: Mistakes corrected.

2.- Abbreviations must be defined when used the first time: MS, TDZ, BAP, NAA, FW, DW, OD… It is necessary to reach section 3.5 for an appropriate description of some of them.

Response: List of abbreviations is added.

3.- Section 2.1. Please, describe in detail all the “high-quality salts, chemicals, and solutions used”.

Response: Changes has been made.

4.- Line 152. HPLC with “standard grade” solvents? In HPLC, highest purity solvents are required.**

Response: This is the case. HPLC section has been revised for clarity.

5.- Line 156 is confusing. Solvent “B” is formic acid or ethanol?

Response: It is methanol. Mistake is corrected in the manuscript.

6.- Line 168: apart of calibration graph and R2 coefficient, please, provide detection and quantitation limits as well as the linear range.**

Response: This HPLC section has been revised accordingly.

7.- Lines 193-194. Please, revise the full sentence.

Response: Changes has been made.

8.- Line 197: form or “from”?

Response: Mistake corrected.

9.- Line 216: Please, change “at 330 nm excitation wavelength and 440 emission wavelengths.” to “at 330 nm (excitation) and 440 (emission) wavelengths.”

Response: Changes has been made.

10.- Lines 243 & 251. What do you mean exactly with “lacking sample extract”?

Response: Lacking sample extract means the solution of standard drug without any sample extract. It has been revised for clarity.

11.- Line 155. Please, revise the sentence. What is the meaning of “treated different”?

Response: Sentence has been revised.

12.- Lines 317-319. Is this an internal commentary of a pre-final version of the manuscript? If so, please delete it.

Response: Yes it is. It has been deleted.

13.- Line 329 and others: Please, make well use of significant figures (13.58889±0.63) -> (13.59±0.63). This revision must be made throughout the manuscript.

Response: Changes has been made.

14.- The meaning of *, ** and *** is not provided in table 1

15.- Line 405: between mean and SD a “±” symbol must be provided.

Response: Mistake has been corrected.

16.- Lines 442-449 should be moved to introduction or deleted, since the information is not related with results. The same can be said for lines 468-483.

Response: Lines are removed.

17.- Conclusions should be condensed. Now they are a mere summary of the paper.

Response: Conclusion section has been revised.

Reviewer 2 Report

The manuscript reports the analysis of bioactive secondary metabolites in callus raised from in vitro leaves explants of Solanum xanthocarpum. A lot of work has been done. However, I found some problems that I point and comment below.

General comments

The title of the manuscript does not reflect the results. At my thought there is no real optimization of tissue culture. The only one type of study was performed (effect of PGRs at different concentrations) using the one explant type (in vitro leaves). In fact, few modifications have been made to talk about optimization of tissue culture. I think authors should modify the title. 

 In terms of plant tissue culture, optimization mean that autors has at least the ‘control’ medium (one of the previously published mediums containing a certain composition and concentration of phytohormones) or any other medium to compare after one of the steps of optimization. In this case, comparisons between various PGRs variants could be made in pairs with control medium or as a Anova test between the all variants when PGRs control is absent (like in the present study).

In the present manuscript Tables 1-6 include ‘control’ that is not the “in vitro tissues control” (medium), but the methodological standard that is used to analyze specific bioactive secondary metabolites. Such interpretation raises questions as the study compares results after cultivation on medium supplemented with various PGRs concentrations. This makes the understanding of results confusing and need modifying and explanation of the term ‘Control’ in Result section and in each table.

Methods

Line 113 – authors need to indicate the number of explants used for each concentration in experiment. This is a very important thing, especially when tissue culture experiments are discussed. 

Lines 124, 130, 131 should be μL instead of μl

Line 135 - Insert bracket

Line 189 – please correct the 300mM, 10mM, 20mM

Line 232 - correct μl

Results and discussion

Line 306-307. TDZ is one of the many used phytohormones in plant tissue culture, but not essential (DOI: 10.5897/AJB11.636), please modify this sentence

Line 307-308. TDZ displays primary cytokinin-like activity, with some auxin-like effects, it can stimulate the accumulation of endogenous auxins in tissues exposed to TDZ (see:  DOI: 10.1007/s00299-007-0357-0, DOI: 10.5897/AJB11.636). Please, modify this part.

Line 317-319. The text ” This section may be divided by subheadings. It should provide a concise and precise description of the experimental results, their interpretation, as well as the experimental conclusions that can be drawn “ should be removed

Line 329 the number 13.58889 should be shortened to 13.59

Line 338. Ullah et al. are mentioned as authors of [21], but in reference list the number 21 corresponds to Velioglu Y., et al., please correct.

Lines 339, 340 and 341, please, indicate the complete names of species.

Line 371 please, indicate the complete name of species.

Line 396. Correct 0.560.021 to 0.56±0.021

Line 432 please, indicate the complete name of species.

Line 436. Add ‘dot’ after [47] 

Tables

 Authors need to make tables more uniform. First of all, the indication of MS0 in Table 1 and Table 2 should be removed, as it was done in Tables 3-6.

Table 1, Line 334 – what is ‘control values’? I cannot find information concerning control medium in the Methods section. The significance of the obtained results for FW and GW is not explained. I recommend using letters (a, b, c, d …) to indicate the critical differences between PGR variants.

Table 2 I advise to change the order of variants of the plant growth regulators, and to place BAP variants in the top of the table before TDZ variants (like in other Tables). Also change PGRS to PGRs, remove MS0 from all lines of table, and replace O.1 TDZ with 0.1 TDZ  

Tables 2 and 3 need the information concerning statistical analysis (Anova results?).

Table 4. replace PGRs Conc. with PGRs concentration (mg/L). I recommend using letters (a, b, c, d …) to indicate the critical differences between the PGR variants. What is control? Is it a standard? Is it also indicated in %, please clarify. In method section (line 179) there is indication of negative control, what kind of ‘control’ is indicated in the footnote of table?

 Table 5. please indicate what is Control in this table. Line 465 indicate that data are compared with control. The elastase inhibition in control is 59.48 (%?), the inhibition in variant 5TDZ+1NAA is 4.12%, according to provided footnotes and table data, there is no differences between 59.48 and 4.12 (market as *); the value of inhibition in the variant 10TDZ+1NAA is 3.32% and the difference with control value of 59.48 is significant (market as **). Maybe it is significant difference between relative values of variants, but not with “control”. Please, explain and clarify. Control is included here in the table, in the Tables 1 and 4 it is out of the Tables. Please correct or explain.  

 Table 6. Same questions as for Table 5 and 4. What is control? There was no ‘Control’ medium in PRGs experiment according to the presented results. The significance between the values is not understood. The values are compared among themselves or with ‘control (which is standard values)’? It should be clarified at in footnotes.

Taking into account the required improvements in the presentation of results in tables, the text in the Results section should also be changed after correcting the tables. 

References

Please, modify the presentation of reference list. Check the Instruction https://www.mdpi.com/authors/references

The complete names of authors of publication should be presented.

Title of the publication should be regular, not italic.

Journal name should be italic. And so on.

Also, please, see the attached version of manuscripts with some minor additional comments and discovered errors.

Author Response

REVIEWER 2

The manuscript reports the analysis of bioactive secondary metabolites in callus raised from in vitro leaves explants of Solanum xanthocarpum. A lot of work has been done. However, I found some problems that I point and comment below.

General comments

The title of the manuscript does not reflect the results. At my thought there is no real optimization of tissue culture. The only one type of study was performed (effect of PGRs at different concentrations) using the one explant type (in vitro leaves). In fact, few modifications have been made to talk about optimization of tissue culture. I think authors should modify the title. 

Response: The title has been revised

 In terms of plant tissue culture, optimization mean that autors has at least the ‘control’ medium (one of the previously published mediums containing a certain composition and concentration of phytohormones) or any other medium to compare after one of the steps of optimization. In this case, comparisons between various PGRs variants could be made in pairs with control medium or as a Anova test between the all variants when PGRs control is absent (like in the present study).

In the present manuscript Tables 1-6 include ‘control’ that is not the “in vitro tissues control” (medium), but the methodological standard that is used to analyze specific bioactive secondary metabolites. Such interpretation raises questions as the study compares results after cultivation on medium supplemented with various PGRs concentrations. This makes the understanding of results confusing and need modifying and explanation of the term ‘Control’ in Result section and in each table.

Response: We have revised it. Standard drugs used as positive controls for each biological activity have been more clearly indicated in Tables footers. For callus, we agree the reviewer point of view and comment, so the term “control” is not used anymore and the statistical analysis revised accordingly.

Methods

Line 113 – authors need to indicate the number of explants used for each concentration in experiment. This is a very important thing, especially when tissue culture experiments are discussed. 

Response: Comment addressed; 5 explants are used for each concentration.

Lines 124, 130, 131 should be μL instead of μl

Response: Changes has been made.

Line 135 - Insert bracket

Response: bracket has been inserted.

Line 189 – please correct the 300mM, 10mM, 20Mm

Response: Mistake corrected.

Line 232 - correct μl

Response: Mistake corrected.

Results and discussion

Line 306-307. TDZ is one of the many used phytohormones in plant tissue culture, but not essential (DOI: 10.5897/AJB11.636), please modify this sentence

Line 307-308. TDZ displays primary cytokinin-like activity, with some auxin-like effects, it can stimulate the accumulation of endogenous auxins in tissues exposed to TDZ (see:  DOI: 10.1007/s00299-007-0357-0, DOI: 10.5897/AJB11.636). Please, modify this part.

Response: Changes has been made.

Line 317-319. The text ” This section may be divided by subheadings. It should provide a concise and precise description of the experimental results, their interpretation, as well as the experimental conclusions that can be drawn “ should be removed

Response: This section has been removed.

Line 329 the number 13.58889 should be shortened to 13.59

Response: Number has been shortened.

Line 338. Ullah et al. are mentioned as authors of [21], but in reference list the number 21 corresponds to Velioglu Y., et al., please correct.

Response: It was typing mistake, now it has been corrected, the correct reference is now included.

Lines 339, 340 and 341, please, indicate the complete names of species.

Response: Complete names of species are mentioned.

Line 371 please, indicate the complete name of species.

Response: Complete names of species are mentioned.

Line 396. Correct 0.560.021 to 0.56±0.021

Response: Mistake corrected.

Line 432 please, indicate the complete name of species.

Response: Name of species indicated.

Line 436. Add ‘dot’ after [47] 

Response: Dot added.

Tables

 Authors need to make tables more uniform. First of all, the indication of MS0 in Table 1 and Table 2 should be removed, as it was done in Tables 3-6.

Response: Comment addressed.

Table 1, Line 334 – what is ‘control values’? I cannot find information concerning control medium in the Methods section. The significance of the obtained results for FW and GW is not explained. I recommend using letters (a, b, c, d …) to indicate the critical differences between PGR variants.

The results of the remaining tables I,e antioxidant activities, anti-inflammatory activities, antiaging activities and antidiabetic activities are compared with respective standard drugs, not control.

Tables 2 and 3 need the information concerning statistical analysis (Anova results?).

Response: We have revised it. Standard drugs used as positive controls for each biological activity have been more clearly indicated in Tables footers. For callus, we agree the reviewer point of view and comment, so the term “control” is not used anymore and the statistical analysis revised accordingly.

Table 2 I advise to change the order of variants of the plant growth regulators, and to place BAP variants in the top of the table before TDZ variants (like in other Tables). Also change PGRS to PGRs, remove MS0 from all lines of table, and replace O.1 TDZ with 0.1 TDZ  

Response: Changes has been made.

Table 4. replace PGRs Conc. with PGRs concentration (mg/L). I recommend using letters (a, b, c, d …) to indicate the critical differences between the PGR variants. What is control? Is it a standard? Is it also indicated in %, please clarify. In method section (line 179) there is indication of negative control, what kind of ‘control’ is indicated in the footnote of table?

 Table 5. please indicate what is Control in this table. Line 465 indicate that data are compared with control. The elastase inhibition in control is 59.48 (%?), the inhibition in variant 5TDZ+1NAA is 4.12%, according to provided footnotes and table data, there is no differences between 59.48 and 4.12 (market as *); the value of inhibition in the variant 10TDZ+1NAA is 3.32% and the difference with control value of 59.48 is significant (market as **). Maybe it is significant difference between relative values of variants, but not with “control”. Please, explain and clarify. Control is included here in the table, in the Tables 1 and 4 it is out of the Tables. Please correct or explain.  

Response: In present manuscript, data of all the biological activities are compared with a standard drug which are mentioned in materials and methods of each activity. We have revised it. Standard drugs used as positive controls for each biological activity have been more clearly indicated in Tables footers.

Table 6. Same questions as for Table 5 and 4. What is control? There was no ‘Control’ medium in PRGs experiment according to the presented results. The significance between the values is not understood. The values are compared among themselves or with ‘control (which is standard values)’? It should be clarified at in footnotes.

Response: Mistakes corrected.

Taking into account the required improvements in the presentation of results in tables, the text in the Results section should also be changed after correcting the tables. 

References

Please, modify the presentation of reference list. Check the Instruction https://www.mdpi.com/authors/references

The complete names of authors of publication should be presented.

Title of the publication should be regular, not italic.

Journal name should be italic. And so on.

 Response: References are now according to mdpi format

Also, please, see the attached version of manuscripts with some minor additional comments and discovered errors.

 Response: Thank you very much.

Round 2

Reviewer 2 Report

I congratulate authors, new version of the manuscript is much better!!

minor corrections:

Line 247 change w/v to italic

Table 1. change the order of the PGR and concentration: TDZ 0.1 to 0.1 TDZ, TDZ 1 to 1 TDZ, TDZ 2.5 to 2.5 TDZ and so on to make this table uniform with the others

Table 2. arrange the order of the lines in PGRs collumn, place variants 0.1 TDZ, 1 TDZ, 2.5 TDZ, 5 TDZ, 10 TDZ at the top of the table, and 0.1 BAP, 1 BAP, 2.5 BAP, 5 BAP, 10 BAP in the middle of table to make this table uniform with the others.

Please check if the references 16, 42, 43 are correctly presented.

In Acknowledgments, please, remove template sentence ‘This section is not mandatory but can be added to the manuscript if the discussion is unusually long or complex’

Author Response

REVIEWER 2 :

I congratulate authors, new version of the manuscript is much better!!

RESPONSE: Thank you very much for your appreciation and your comments and suggestions that greatly help us in the revision of the ms. The corrections appear in red in the revised version.

minor corrections:

Line 247 change w/v to italic

RESPONSE: It has been corrected.

Table 1. change the order of the PGR and concentration: TDZ 0.1 to 0.1 TDZ, TDZ 1 to 1 TDZ, TDZ 2.5 to 2.5 TDZ and so on to make this table uniform with the others

RESPONSE: Thank you very much for your suggestion. The modification has been done.

Table 2. arrange the order of the lines in PGRs collumn, place variants 0.1 TDZ, 1 TDZ, 2.5 TDZ, 5 TDZ, 10 TDZ at the top of the table, and 0.1 BAP, 1 BAP, 2.5 BAP, 5 BAP, 10 BAP in the middle of table to make this table uniform with the others.

RESPONSE: Thank you. It has corrected accordingly.

Please check if the references 16, 42, 43 are correctly presented.

RESPONSE: Thank you. These references have been reformatted.

In Acknowledgments, please, remove template sentence ‘This section is not mandatory but can be added to the manuscript if the discussion is unusually long or complex’

RESPONSE: It has been deleted.
